# Counterfactual Explanations for Optimization-Based Decisions in the Context of the GDPR

**Anton Korikov** , **Alexander Shleyfman** and **J. Christopher Beck**

Department of Mechanical & Industrial Engineering, University of Toronto, Toronto, Canada

{korikov, jcb}@mie.utoronto.ca, shleyfman.alexander@gmail.com

## Abstract

The General Data Protection Regulations (GDPR) entitle individuals to explanations for automated decisions. The form, comprehensibility, and even existence of such explanations remain open problems, investigated as part of explainable AI. We adopt the approach of counterfactual explanations and apply it to decisions made by declarative optimization models. We argue that inverse combinatorial optimization is particularly suited for counterfactual explanations but that the computational difficulties and relatively nascent literature make its application a challenge. To make progress, we address the case of counterfactual explanations that isolate the minimal differences for an individual. We show that under two common optimization functions, full inverse optimization is unnecessary. In particular, we show that for functions of the form of the sum of weighted binary variables, which includes frameworks such as weighted MaxSAT, a solution can be found by solving a slightly modified version of the original optimization model. In contrast, the sum of weighted integer variables can be solved with a binary search over a series of modifications to the original model.

## 1 Introduction

There has been substantial development of algorithmic systems that make recommendations or decisions about high-impact issues such as healthcare, homelessness, recidivism and parole, and social services [Eubanks, 2018; Azizi *et al.*, 2018]. The General Data Protection Regulations (GDPR) [Parliament and Council of the European Union, 2016] declare that individuals subjected to an automated decision have a right to an explanation of the decision. While the legal scope of such a right is an active topic of debate, there is widespread consensus on the importance of implementing such a concept [Selbst and Powles, 2018; Wachter *et al.*, 2017a] and research on explanations is a central direction in the field of explainable AI (XAI) [Guidotti *et al.*, 2018].

For example, consider an optimization-based automated medical scheduling system that assigns an individual to an appointment three months in the future. The individual may ask a question such as "Why can't I be seen next month?"

In this paper, we adopt the approach of *counterfactual explanations* [Wachter *et al.*, 2017b] by deriving the minimal change to the world that would have resulted in the alternative indicated by the questioner. We develop an approach to generate counterfactual explanations for optimization-based decisions using a generalization of inverse combinatorial optimization [Heuberger, 2004]. Under some restrictions, we show how to formulate the inverse problem and, further, how to solve it under two common objective functions: the sum of weighted binary variables and the sum of weighted integer variables. Our solution techniques demonstrate that the computational effort to produce an explanation is equivalent to that of the original optimization problem in the former case and a log-factor more expensive than that in the latter case.

In the next section, we provide background on counterfactual explanations and inverse optimization before turning to our formal problem definition in Section 3. Sections 4 and 5 present our contributions w.r.t. the two objective functions and we then discuss the limitation of our work and relate it to existing work in the final sections.

## 2 Background

### 2.1 Counterfactual Explanations in XAI

A large part of XAI, especially in machine learning (ML), has focused on explaining decision by making the internal logic of an algorithm more understandable [Burrell, 2016]. A common approach is to generate local approximations of opaque models (e.g., neural networks) with interpretable models (e.g., linear models or decision trees) from which the impact of features on decisions is easier to comprehend [Ribeiro *et al.*, 2016; Simonyan *et al.*, 2013]. This approach has been criticized as insufficient in generating the explanations envisioned by the GDPR, since it does not explicitly describe the circumstances under which a different decision would have been reached. Such a lack makes it difficult to *contest* the decision and to *act* to change it, two key goals for user-centered explanations [Wachter *et al.*, 2017b].

A different line of research in XAI addresses these issues with *counterfactual explanations* [Wachter *et al.*, 2017b] which have the following form: given that a set of facts resulted in decision $x$, decision $\bar{x}$ would have been made had

these facts been changed in a particular way and the rest of the world had remained unchanged. Using our appointment scheduling example: the individual's medical acuity level was 3 and the appointment was scheduled in three months; had the acuity level been 5 (i.e., more acute), the appointment would have been scheduled next month.

More precisely, a counterfactual is situation that is "contrary to the facts": an imagined alternative to a past event. Counterfactual reasoning invokes a comparison between the imagined event and the *factual* event [Epstude and Roese, 2008]. By this definition, a counterfactual explanation involves two counterfactuals. First there is a *counterfactual question*, "Why decision $x$ and not $\bar{x}$?", with the imagined decision $\bar{x}$ being the first counterfactual. The explanation then provides a set of hypothetical facts, different from the original facts, as the second counterfactual. To distinguish between the two counterfactuals, we refer to decision $\bar{x}$ as the *foil* and the hypothetical facts as the *counterfactual values*, using similar terminology to Miller [2019]. Further, it is typically preferable for such an explanation to minimally perturb the facts that led to the original decision.

As a counterfactual explanation explicitly addresses what needs to happen for a decision to be different and avoids the internal logic of the algorithm, such an explanation is better suited for explanations for users [Wachter *et al.*, 2017b].

## 2.2 Inverse Combinatorial Optimization

Standard (or, *forward*) optimization searches for an optimal assignment of variables given a model consisting of an objective function, constraints, and parameters. Inverse optimization, in contrast, starts with a model, initial parameter values, and a known solution, and finds the minimal change to the parameter values that would make the solution optimal [Demange and Monnot, 2014; Heuberger, 2004].

To formulate the forward and inverse problems, let $x \in X \subseteq \mathbb{R}^n$ be a decision vector subject to a feasible set $X$, $c \in \mathcal{C} \subseteq \mathbb{R}^n$ be an initial parameter vector subject to feasible set $\mathcal{C}$, and $f : \mathcal{C} \times X \to \mathbb{R}$ be the objective function. It is assumed that the components of $c$ are present *only* in the objective, not in any constraints. The minimization variant of the forward problem $\langle X, c, f \rangle$ is to find an optimal $x^*$ such that $f(c, x^*) = \min\{f(c, x) : x \in X\}$.

In the inverse problem, defined w.r.t. a norm $\|\cdot\|$, we are given a solution $\hat{x} \in X$ that we wish to make optimal by minimally modifying the parameter vector. Let $\bar{c} \in \mathcal{C}$ be the modified parameter vector. The inverse optimization problem is then

$$\min \|\bar{c} - c\| \tag{1}$$
$$\text{s.t. } f(\bar{c}, \hat{x}) = \min\{f(\bar{c}, x) : x \in X\} \tag{2}$$
$$\hat{x} \in X, \quad \bar{c} \in \mathcal{C} \tag{3}$$

In *partial inverse optimization* [Demange and Monnot, 2014], we are no longer given a full solution $\hat{x}$, but instead a partial solution $\bar{x}_i^p \in \mathbb{R}$ for $i \in I \subseteq \{1, \ldots, n\}$ and, similarly, must find minimally perturbed parameters. The partial inverse optimization problem is identical to the above model with the additional constraint: $\hat{x}_i = \bar{x}_i^p, \forall i \in I$.

## 3 Problem Definition

Let $x^*$ be an optimal solution to an instance of a (forward) optimization problem, $\langle X, c, f \rangle$. Assume we are asked a counterfactual question "Why is $x^*$ not different?" such that this difference can be expressed by a *foil constraint set*, $\psi$, that $x^*$ does not satisfy. In our medical scheduling example, this question might be "Why isn't my appointment scheduled next month?", and $\psi$ would represent the constraint that the patient's procedure is in the next month.

Let $X_\psi \subseteq X$ represent all points in $X$ that satisfy $\psi$. Any point in $X_\psi$ is a *foil* (as per Section 2.1), because it represents a counterfactual decision the user has asked about. We assume $X_\psi \neq \emptyset$. Let $X_\psi^c = X \setminus X_\psi$ be the complement of $X_\psi$, and any point in $X_\psi^c$ be called a *factual* solution (by definition, $x^* \in X_\psi^c$). An explanation is then generated by finding the counterfactual parameters $\bar{c} \in \mathcal{C}$ that cause a foil to become optimal while minimally perturbing the initial parameters $c$. We therefore define the Nearest Counterfactual Explanation (NCE) problem for some norm $\|\cdot\|$ as:

$$\min_{\bar{c} \in \mathcal{C}} \|\bar{c} - c\| \tag{4}$$
$$\text{s.t. } \min_{x \in X} f(\bar{c}, x) = \min_{x \in X_\psi} f(\bar{c}, x). \tag{5}$$

The explanation, then, is: "If the parameters had been $\bar{c}$ instead of $c$, the alternative decision would have been reached."

This formulation is a novel generalization of both inverse optimization and partial inverse optimization, since it allows a foil to be described with a constraint set, not only a (partial) set of value assignments. For example, a patient can ask why they are not seen in a given time interval, as opposed to only asking why they are not seen on a given day.

### 3.1 Restrictions

General inverse optimization is computationally difficult, especially with discrete variables [Demange and Monnot, 2014], and the literature is still developing robust solution approaches. We make the following restrictions here to make progress on a subspace that nonetheless includes a number of interesting problems, including our medical example.

1. We limit $\psi$ to one constraint on one component of $x$ with the form $\psi : x_j \leq m_0$ or $\psi : x_j \geq m_0$, $j \in [n]$, $m_0 \in \mathbb{N}_0$.[1] We are interested in explanations for one individual, often modeled with a single variable and parameter, $x_j$ and $c_j$, and our explanations consider only changes to $c_j$, and not the other components of $c$. This restriction means that an explanation is about a characteristic of the questioner as opposed to others that the questioner may have no knowledge of. It also protects the privacy of others as their real and counterfactual parameter values are not revealed in an explanation.

2. For the original parameters, $c$, all foils are strictly suboptimal solutions, i.e. $\nexists x \in X_\psi$ such that $\min\{f(c, x) : x \in X\} = \min\{f(c, x) : x \in X_\psi\}$. We do not address cases where the foil is an alternative optimal solution to the original problem because such a situation produces

---

[1] We use $[n]$ to denote the interval of values $\{1, \ldots, n\}$.

the unsatisfying explanation: "The foil is just as suitable as the initial decision, but was not selected due to arbitrary details of the algorithm (e.g., tie breaking)."

3. The components of $c$ and $\bar{c}$ appear only in the objective function, not in any constraints. Therefore, any $x \in X$ remains feasible for all values of $c$ and $\bar{c}$.

4. There exists at least one feasible foil: $X_\psi \neq \emptyset$.

We return to these restrictions in more depth in Section 6. We now address the NCE problem for two common objective functions: the weighed sum of binary variables and the weighted sum of general integer variables.

## 4 Weighted Sum of Binary Variables

Consider a problem where a subset of individuals are chosen from a group, for example, to receive some service. For such a problem, a natural model is to represent each individual as a binary variable where the 1 represents provision of the service and where some individual measure of the importance of each individual is represented by a scalar value. The appropriate objective function is to maximize the sum of the quality of the chosen individuals and the obvious foil question from an individual is "Why wasn't I selected for the service?"

**Definition 4.1** (Weighted Sum of Binary Variables (WSBV)). *Let $x \in X \subseteq \{0,1\}^n$ be a binary decision vector, $c \in \mathbb{N}_0^n$ a vector of integer parameters such that $c$ is not present in any constraints that define $X$, and $c \cdot x = \sum_{i=1}^n c_i x_i$ be a scalar product of two vectors in $\mathbb{R}^n$.[2] The WSBV maximization problem is $\max\{c \cdot x : x \in X\}$.*

Let $NCE_{0\text{-}1}$ be the NCE problem for WSBV. We are given a WSBV instance $\langle X, c \rangle$ and a foil constraint $\psi$. Because $x$ is a binary vector and due to restriction 1, $\psi$ must have the form $\psi : x_j = 1$ or $\psi : x_j = 0$. Let $\bar{c} \in \mathbb{N}_0^n$ be the modified parameter vector, which is identical to $c$ with the exception of component $\bar{c}_j$ replacing $c_j$, since we are only interested in that one parameter. Thus, using the $L_1$ norm, we define the $NCE_{0\text{-}1}$ problem as

$$\min_{\bar{c}_j \geq 0} |\bar{c}_j - c_j| \tag{6}$$

$$\text{s.t.} \max_{x \in X} \bar{c} \cdot x = \max_{x \in X_\psi} \bar{c} \cdot x. \tag{7}$$

$$\bar{c}_i = c_i, \quad \forall i \in [n], \quad i \neq j \tag{8}$$

Rather than solving a general inverse optimization problem, we can solve this problem in closed-form given optimal solutions to the original problem and to the original problem plus $\psi$. Intuitively, any reduction in the objective function value incurred by the foil must be exactly compensated by the increase in $\bar{c}_j$ in order for the foil to be optimal.

Let the following function denote the difference between the optimal foil and optimal factual solution

$$\Delta_\psi^X(y) = \max_{x \in X} y \cdot x - \max_{x \in X_\psi} y \cdot x.$$

Since $X_\psi \subseteq X$, it holds that $\Delta_\psi^X(y) \geq 0$ for any $\psi$, $X \subseteq \{0,1\}^n$, and $y \in \mathbb{N}_0^n$. We prove the validity of our closed

form approach for the case when $\psi : x_j = 1$, but it can easily be modified for the case of $\psi : x_j = 0$, as well as the minimization variant of a WSBV problem.

**Theorem 1.** *The $\langle X, c, \psi \rangle$ $NCE_{0\text{-}1}$ problem, with $\psi : x_j = 1$ and a non-empty foil set $X_\psi \neq 0$ has an optimal solution $\bar{c}_j^* = c_j + \Delta_\psi^X(c)$.*

*Proof.* By restriction 2, none of the optimal solutions of $\langle X, c \rangle$ satisfy $\psi$.[3] In this case we can write

$$\Delta_\psi^X(c) = \max_{x \in X_\psi^c} c \cdot x - \max_{x \in X_\psi} c \cdot x.$$

Since the foil constraint has the form $\psi : x_j = 1$, for each $x \in X_\psi^c$ we have $x_j = 0$. Recalling that $\bar{c}$ is identical to $c$ everywhere except the coordinate $j$, for every $\bar{c}$ we have that

$$\max_{x \in X_\psi^c} c \cdot x = \max_{x \in X_\psi^c} \bar{c} \cdot x.$$

Similarly, for each $x \in X_\psi$ we must have $x_j = 1$ and that the contribution of the non-$j$ components must be identical for every $\bar{c}$, such that

$$\max_{x \in X_\psi} c \cdot x - c_j = \max_{x \in X_\psi} \bar{c} \cdot x - \bar{c}_j$$

Next, note that by (7) we have that for every solution $\bar{c}$ of an NCE $\langle X, c, \psi \rangle$ problem, it holds that $\Delta_\psi^X(\bar{c}) = 0$. Thus,

$$0 = \Delta_\psi^X(\bar{c}) = \max_{x \in X_\psi^c} \bar{c} \cdot x - \max_{x \in X_\psi} \bar{c} \cdot x =$$

$$\max_{x \in X_\psi^c} c \cdot x - \max_{x \in X_\psi} c \cdot x + c_j - \bar{c}_j =$$

$$\Delta_\psi^X(c) + c_j - \bar{c}_j.$$

Therefore, we have $\bar{c}_j = c_j + \Delta_\psi^X(c)$. It is clear that $\bar{c}_j \in \mathbb{N}_0$, because both $c \in \mathbb{N}_0$ and $x \in \mathbb{N}_0$, and therefore $\Delta_\psi^X(c) \in \mathbb{N}_0$. Optimality follows from uniqueness of $\bar{c}$. $\square$

Theorem 1 shows that the computational effort required to form an explanation arises solely from re-solving the original problem with the added constraint $x_j = 1$. As mentioned, the proof can easily be modified when the foil constraint is $\psi : x_j = 0$, to give the optimal solution $\bar{c}_j^* = c_j - \Delta_\psi^X(c)$.

**The Scope of Explanations for WSBV** We now show two examples of explanations that can be generated as a result of solving $NCE_{0\text{-}1}$. In the 0-1 knapsack problem (KP) we are given a knapsack capacity $W \in \mathbb{N}$ and a set of items $[n] \subseteq \mathbb{N}$, with each item $i \in [n]$ associated with a weight $w_i \in \mathbb{N}$, a profit $c_i \in \mathbb{N}$, and a binary variable $x_i \in \{0,1\}$ set to 1 iff the item is included in the knapsack. The 0-1 KP is $\max\{\sum_{i=1}^n x_i c_i : \sum_{i=1}^n x_i w_i \leq W\}$, that is, to select a subset of items such that the sum of the profits of the selected items is maximized and the sum of their weights does not exceed the knapsack capacity.

KP problems are used for many real-life applications, for instance for capital budgeting decisions and our example of providing a limited service [Kellerer *et al.*, 2013]. Using the

---

[2]Since it is obvious from the context, we write $c \cdot x$ instead of $c^T \cdot x$ to avoid multiple superscript indices.

[3]The theorem is also true, if somewhat trivially, without restriction 2, by setting $\bar{c}_j^* = c_j$.

NCE$_{0\text{-}1}$, the explanation for a person not receiving a service can be "Because the benefit of offering you this service was only $c_j$. If it had increased to $\bar{c}_j^*$, you would have been offered the service." Conversely, a person who *was* offered the service can assess how much their benefit $c_j$ can decrease before the service is assigned to someone else.

Another important class of WSBV problems is the weighted MaxSAT problem [Li and Manya, 2009]. We are given a conjunction of $n$ disjunctive clauses, with each clause $i \in [n]$ associated with a weight $c_i \in \mathbb{N}$ and a binary variable $x_i \in \{0, 1\}$ that is assigned 1 iff the clause is satisfied. The objective is to maximize the total weight of the satisfied clauses, $\max \sum_{i=1}^{n} x_i c_i$. Using our explanation technique, we can ask "Why is clause $j$ not satisfied?", and receive an explanation "Because the weight of this clause was only $c_j$. If it increased to $\bar{c}_j^*$, it would be satisfied."

# 5 Weighted Sum of Integer Variables

We now consider a problem class that includes the medical appointment scheduling example introduced above. In such a problem, the goal is to schedule appointments to maximize some form of health outcome which is often approximated by the minimization of the acuity-weighted sum of appointment times. A foil question from an individual would have the form of asking why their appointment was not scheduled at some other time (e.g., earlier).

**Definition 5.1** (Weighted Sum of Integer Variables (WSIV)). *Let $x \in X \subseteq \mathbb{N}_0^n$ be an integer decision vector and $c \in \mathbb{N}_0^n$ an integer parameter vector, such that $c$ is not present in the constraints that define $X$. A WSIV minimization problem is of the form $\min\{c \cdot x : x \in X\}$.*

We define the explanation problem NCE$_{\mathbb{N}_0}$. Given is a WSIV instance $\langle X, c \rangle$ and a foil constraint $\psi$, which by restriction 2 takes the form $\psi : x_j \leq m_0$ or $\psi : x_j \geq m_0$, $m_0 \in \mathbb{N}_0$, over the coordinate $j \in [n]$ of the variables $x$. As before, let $\bar{c}$ be the modified parameter vector, identical to $c$ for all coordinates other than $j$. The NCE$_{\mathbb{N}_0}$ using the $L_1$ norm is then

$$\min_{\bar{c}_j \geq 0} |\bar{c}_j - c_j| \tag{9}$$

$$\text{s.t.} \min_{x \in X} \bar{c} \cdot x = \min_{x \in X_\psi} \bar{c} \cdot x. \tag{10}$$

$$\bar{c}_i = c_i, \quad \forall i \in [n], \quad i \neq j \tag{11}$$

In our appointment scheduling example, we are given $n$ patients (jobs) and $m$ medical resources. Let $c \in \mathbb{N}_0^n$ represent patient acuity (a higher value means higher acuity), $p \in \mathbb{N}_0^n$ represent job processing times, and $x \in X \subseteq N_0^n$ represent job completion times, where $X$ contains any constraints on the schedule (e.g., resource capacity, task precedence, etc.). The goal is to find a schedule that minimizes the weighted sum of completion times, $\min\{c \cdot x : x \in X\}$. The question "Why can't I be seen before time $m_0$?" is exactly represented by the NCE$_{\mathbb{N}_0}$ problem with $\psi : x_j \leq m_0$.

## 5.1 Theoretical Results

We present theoretical results for the $\psi : x_j \leq m_0$ case. The proof can be modified to show that Theorem 2 holds for the case of $\psi : x_j \geq m_0$ as well.

The NCE$_{\mathbb{N}_0}$ is more involved than NCE$_{0\text{-}1}$ because we cannot rely on the fact that $x_j = 0$ in the optimal factual solution. As a result, modifications to the $c_j$ parameter change the objective value of both the optimal counterfactual and factual solutions and we must reason about the change in their relative values as $c_j$ varies.

We aim to prove that there is a constant $c_j^* \in \mathbb{R}_{0+}$ such that $[c_j^*, \infty) \cap \mathbb{N}_0$ is the set of all feasible solutions to the NCE, i.e. all values $\bar{c}_j \in \mathbb{N}_0$ result in feasible solutions iff $\bar{c}_j \geq c_j^*$. This result will allow the use of binary search over a number of instances of the forward optimization problem to solve the NCE. To this end, as above, we define an auxiliary function

$$\Delta_\psi^X(y) = \min_{x \in X_\psi} y \cdot x - \min_{x \in X} y \cdot x.$$

To establish the observation above it is enough to prove the following.

**Theorem 2.** *Let $\langle X, c, \psi \rangle$ be an NCE problem with $\psi : x_j \leq m_0$. Then,*

1. *$\Delta_\psi^X(y)$ is a monotonically non-increasing function in $y_j$.*

2. *There is a $c_j^0 \in \mathbb{R}_{0+}$ such that $\Delta_\psi^X(c^0) = 0$ and $c^0$ coincides with $c$ on every entry other than $j$.*

In what follows, we assume $y$ and $\bar{y}$ are two vectors in $\mathbb{R}_{0+}^n$ that are identical over all coordinates except $j$ where $y_j \leq \bar{y}_j$, and denote their positive difference by $\delta_y$. Intuitively, we show that if $y_j$ increases by positive $\delta_y$, both components of $\Delta_\psi^X(\bar{y})$ will increase, but the negative component $\min_{x \in X} \bar{y} \cdot x$ will increase faster, and the function will eventually vanish.

Let us start with an observation that uses the form of the foil constraint to provide bounds on how fast the components of $\Delta_\psi^X(\bar{y})$ can increase w.r.t. $\bar{y}_j$. Intuitively, part 1 below uses the minimum value of $z_j$ to derive a lower bound on the increase of the optimal factual solution w.r.t. $\bar{y}_j$. Similarly, part 2 uses the maximum value of $z_j$ to give an upper bound on the increase of the optimal foil solution w.r.t. $y_j$.

**Lemma 1.** *Let $Z$ be a finite feasible set.*

1. *If for each $z \in Z$ it holds that $k_0 \leq z_j$, we have $\delta_y k_0 \leq \min_{z \in Z} \bar{y} \cdot z - \min_{z \in Z} y \cdot z$.*

2. *If for each $z \in Z$ it holds that $z_j \leq K_0$, we have $\min_{z \in Z} \bar{y} \cdot z - \min_{z \in Z} y \cdot z \leq \delta_y K_0$.*

*Proof.* Let $z^* \in Z$ be an optimal solution for the minimization problem $\langle Z, y \rangle$, and let $\bar{z}^* \in Z$ be an optimal solution for the minimization problem $\langle Z, \bar{y} \rangle$. Note also that by definition of $\bar{y}$, for each vector $x \in \mathbb{R}^n$ we have

$$\bar{y} \cdot x = y \cdot x + \delta_y x_j. \tag{12}$$

Statement (1): since $z^*$ is optimal given $y$, we have $y \cdot z^* \leq y \cdot \bar{z}^*$. Thus, using (12) we deduce

$$\min_{z \in Z} \bar{y} \cdot z - \min_{z \in Z} y \cdot z \geq \bar{y} \cdot \bar{z}^* - y \cdot \bar{z}^* = \delta_y \bar{z}_j^* \geq \delta_y k_0.$$

Statement (2) is proved similarly. As $\bar{z}^*$ is optimal for $\bar{y}$,

$$\bar{y} \cdot \bar{z}^* \leq \bar{y} \cdot z^* = y \cdot z^* + \delta_y z_j^* \leq y \cdot z^* + \delta_y K_0$$

Then, again using (12) we can write,

$$\min_{z \in Z} \bar{y} \cdot z - \min_{z \in Z} y \cdot z \leq y \cdot z^* + \delta_y K_0 - \min_{z \in Z} y \cdot z \leq \delta_y K_0.$$

□

We can now proceed to the proof of Theorem 2.

*Proof.* Assume that $X_\psi^c \neq \emptyset$, since otherwise both claims hold trivially. First, we aim at proving that $\Delta_\psi^X$ is monotonic non-increasing within the scope of one coordinate. Recall that $y$ and $\bar{y}$ are two vectors in $\mathbb{R}_{0+}^n$ that are identical over all coordinates except $j$ where $y_j \leq \bar{y}_j$, with their positive difference denoted $\delta_y$. We shall prove that

$$\Delta_\psi^X(y) \geq \Delta_\psi^X(\bar{y}).$$

Let us look at the three following cases:

a) The case $\Delta_\psi^X(\bar{y}) = 0$ is trivial, since by definition $\Delta_\psi^X(y) \geq 0$ for every $y \in \mathbb{R}_{0+}^n$.

b) Let $\Delta_\psi^X(y) = 0$. Let $x^{*,\psi} \in X_\psi$ be an optimal solution for the minimization problem $\langle X, y \rangle$. By definition of $\psi$, this means $x_j^{*,\psi} \leq m_0$. Let $x^c \in X_\psi^c$ be a solution in the complement of $X_\psi$, meaning that, $x_j^{*,\psi} \leq m_0 < x_j^c$. In order for $\Delta_\psi^X(y) = 0$, we must have

$y \cdot x^{*,\psi} \leq y \cdot x^c \implies$

$\bar{y} \cdot x^{*,\psi} = y \cdot x^{*,\psi} + \delta_y x_j^{*,\psi} \leq y \cdot x^c + \delta_y x_j^c = \bar{y} \cdot x^c.$

By this, we have that $\min_{x \in X_\psi} \bar{y} \cdot x \leq \min_{x \in X_\psi^c} \bar{y} \cdot x$. Which implies that $\Delta_\psi^X(\bar{y}) = 0$.

c) Lastly, assume

$$\min\{\Delta_\psi^X(y), \Delta_\psi^X(\bar{y})\} > 0 \quad (13)$$

For $\Delta_\psi^X$ to be monotonically non-increasing, the following difference of deltas must be non-positive:

$\Delta_\psi^X(\bar{y}) - \Delta_\psi^X(y) =$

$\min_{x \in X_\psi} \bar{y} \cdot x - \min_{x \in X} \bar{y} \cdot x - \min_{x \in X_\psi} y \cdot x + \min_{x \in X} y \cdot x \quad (14)$

We can observe that (13) will hold iff all foils are suboptimal, meaning that all optimal solutions lie in $X_\psi^c$. Using this condition and rearranging (14),

$\Delta_\psi^X(\bar{y}) - \Delta_\psi^X(y) =$

$\min_{x \in X_\psi} \bar{y} \cdot x - \min_{x \in X_\psi^c} y \cdot x - \min_{x \in X_\psi^c} \bar{y} \cdot x + \min_{x \in X_\psi^c} y \cdot x.$

Next, we insert bounds on these terms from Lemma 1. Note that for each $x \in X_\psi$ it holds that $x_j \leq m_0$. Thus, by Lemma 1 part 2 we have

$$\min_{x \in X_\psi} \bar{y} \cdot x - \min_{x \in X_\psi} y \cdot x \leq \delta_y m_0.$$

Similarly, for each $x \in X_\psi^c$ it holds that $x_j \geq m_0 + 1$. Thus, by Lemma 1 part 1 we have

$$\min_{x \in X_\psi^c} \bar{y} \cdot x - \min_{x \in X_\psi^c} y \cdot x \geq \delta_y m_0 + \delta_y.$$

Applying these two inequalities to 14 we have

$$\Delta_\psi^X(\bar{y}) - \Delta_\psi^X(y) \leq \delta_y m_0 - \delta_y m_0 - \delta_y \leq 0 \implies$$
$$\Delta_\psi^X(\bar{y}) \leq \Delta_\psi^X(y).$$

Next, we prove the second part of Theorem 2, that there is a $c_j^0 \in \mathbb{R}_{0+}$ such that $\Delta_\psi^X(c^0) = 0$ and $c^0$ coincides with the unmodified parameter vector $c$ on every entry other than $j$. We must show that

$$\min_{x \in X_\psi} c^0 \cdot x \leq \min_{x \in X_\psi^c} c^0 \cdot x. \quad (15)$$

Let us define a vector $x^{\min} \in \mathbb{N}_0^n$, where every entry other than $j$ is zero and $x_j^{\min} = m_0 + 1$. For every $x \in X_\psi^c$, we observe that $x_i^{\min} \leq x_i$ for each coordinate $i$. Thus, for every $c' \in \mathbb{R}_{0+}^n$ it holds that $c' \cdot x^{\min} \leq \min\{c' \cdot x : x \in X_\psi^c\}$. Therefore, (15) will hold if we can satisfy

$$\min_{x \in X_\psi} c^0 \cdot x \leq c^0 \cdot x^{\min} = c_j^0(m_0 + 1) \quad (16)$$

To this end, we pick an arbitrary $x^\psi \in X_\psi$ and recall that $x_j^\psi \leq m_0$. We will satisfy (16) if we can find a $c_j^0 \in \mathbb{R}_{0+}$ such that

$$c^0 \cdot x^\psi \leq m_0 c_j^0 + \sum_{i \neq j} c_i x_i^\psi \leq c_j^0(m_0 + 1).$$

Thus, we can pick $c_j^0 = \sum_{i \neq j} c_i x_i^\psi$. □

Note that selecting this value of $c_j^0$ will also provide an upper bound for the binary search, as discussed next.

## 5.2 Solving NCE$_{\mathbb{N}_0}$ Problems

We use the results of Theorem 2 to show that NCE$_{\mathbb{N}_0}$ can be solved with binary search. Theorem 2 assures us that feasible solutions for NCE$_{\mathbb{N}_0}$ $\langle X, c, \psi \rangle$, with $\psi : x_j \leq m_0$, are the integers that lie within the interval $[c_j^*, \infty) \subseteq \mathbb{R}_{0+}$ (i.e. $\Delta_\psi^X$ vanishes for each integer in this interval). Since we are looking for the minimal solution, to solve the problem it is sufficient to find $\lceil c_j^* \rceil$.

The last part of the proof of Theorem 2 provides us with an upper bound on the optimal solution, $c_j^0 \in \mathbb{R}_{0+}$ such that $c_j^* \leq c_j^0$. To compute $c_j^0$, we must find a feasible foil, $x^\psi \in X_\psi$. The smallest possible value of $c_j^0$ will be given by $x^\psi \in \text{argmin}\{c \cdot x - c_j x_j : x \in X_\psi\}$. Alternatively, it may be more convenient to use the original forward objective function, so that $x^\psi \in \text{argmin}\{c \cdot x : x \in X_\psi\}$, though this may result in a higher value for $c_j^0$. The lower bound on $\lceil c_j^* \rceil$ is $c_j + 1$, because $c_j < \lceil c_j^* \rceil$ since all foils are suboptimal at $c_j$ (by restriction 2). Since $\Delta_\psi^X$ is a monotonically non-increasing function, it is enough to run binary search on the integer set $\{c_j + 1, \ldots, \lceil c_j^0 \rceil\}$ to solve the problem. This, together with the identity $\lceil \log_2 \lceil x \rceil \rceil = \lceil \log_2 x \rceil$ for each $x \geq 1$, results in the following statement:

**Proposition 1.** *Let $\langle X, c, \psi \rangle$ be an NCE problem. Given a vector $c^0 \in \mathbb{N}^n$, where $c_j^0$ replaces the $j$'th entry in the vector $c$, such that $\Delta_\psi^X(c^0) = 0$, the $\langle X, c, \psi \rangle$ problem can be solved within at most $\lceil \log_2(c_j^0 - c_j) \rceil$ solutions of the forward WSIV minimization problems of the form $\langle \bar{c}, X_\psi \rangle$ and $\langle \bar{c}, X_\psi^c \rangle$.*

## 6 Discussion and Limitations

We demonstrated that we can generate counterfactual explanations for general optimization models with two common objective functions. The additional computational effort is solving either a single or logarithmic number of forward optimization problems, depending on the objective function. Our contributions were made within a restricted context.

Our first limitation (Section 3.1) restricted the scope for explanation to a single model parameter. In the context of the GDPR, the restriction to characteristics of the questioner is reasonable both for the need to protect private information and from the perspective of generating a meaningful explanation: a counterfactual that changes characteristics of other individuals has limited value. Lifting the restriction to address a set of parameters concerning the questioner would require general partial inverse optimization. While most inverse optimization literature deals with multiple parameters [Heuberger, 2004], especially in the case of discrete variables, such problems are still challenging. We see this extension as a prime area for near-term future research.

Our second and fourth restrictions require that a foil solution exists and is not optimal for the factual problem. Both restrictions were made because it is unclear what kind of counterfactual explanation is appropriate if they are lifted. If the foil solution does not exist, then the Nearest Counterfactual Explanation problem is infeasible and no possible modification can result in the questioner's counterfactual. If the foil solution is optimal for the original problem, then there is no reason why it was not chosen as the factual solution. In either case, there does not appear to be a meaningful explanation.

Finally, we required that none of the counterfactual parameters appear in the constraints. Without such a restriction, not all solutions $x \in X$ would remain feasible as the parameters change, substantially complicating the mathematical structure used in our results. While there exists some work in inverse optimization under such assumptions (e.g., [Yang *et al.*, 1997]), approaches to the problem are not well-developed and form another important research direction.

In addition to the above restrictions, our work relies on the assumption that the users know and understand their weight. For a medical scheduling example, we assume patients know or can access their acuity scores. For the example of solving a knapsack problem to select clients, we assume clients know how much they are ready to pay for a product.

In our framework, a user has several choices after an explanation: they can accept the decision, act to change it (e.g., get a second opinion of their acuity), or contest the decision on the grounds that it is unfair to use this weight in this way. In the latter case, they may ask why the system makes distinctions based on the weight, leading to ethical discussions which are important but beyond the scope of explaining the decision. Though, we note that our approach provides an explanation that can enable a user to pose such meta-explanation questions.

If our assumption does not hold and the user does not understand how their weight was assigned, then they no longer need an explanation of the optimization-based decision, but rather an explanation of how the inputs to the model are determined. This is again beyond our scope as it now demands an explanation from the automated or human system that assigned the weights. But again, by explaining that a change in weight would change the outcome, our approach allows the user to question another part of the decision making system.

## 7 Related Work

Much recent work has looked at counterfactual explanations for classifiers, with seminal work by Wachter et. al [2017b]. Most relevant is work that generates explanations with a binary search over a sequence of combinatorial models [Karimi *et al.*, 2020; Russell, 2019]. To our knowledge, none of the work addresses decisions made by explicit optimization models nor makes the connection to inverse optimization.

Explainable constraint programming has mainly focused on explaining infeasibility [Freuder, 2017], often through the identification of a minimal set of unsatisfiable constraints [Junker, 2004]. A parallel literature exists in mathematical programming [Chinneck, 2008]. While such sets could be viewed as counterfactuals (i.e., one of these constraints must be different), this connection has not been clearly developed and the use of inverse optimization is absent.

Other research has looked at counterfactual explanations in AI Planning, highlighting the difference between specific factual and counterfactual plans [Fox *et al.*, 2017] and describing properties of all counterfactual plans [Eiffer *et al.*, 2020].

The only previous work that we are aware of that relates counterfactual explanations with inverse optimization is that of Brandao and Magazzeni [2020] which generates counterfactual explanations for path planning using an inverse shortest path formulation. While clearly making the connection with inverse optimization, the work focuses on the polynomially solvable inverse shortest path problem and has a substantially narrower scope than ours.

## 8 Conclusion

We have shown that counterfactual explanations for decisions made by an optimization model can be generated through a generalization of inverse combinatorial optimization, a problem that has been studied in mathematical programming. Within the context of legislation such as the General Data Protection Regulations that mandates the right to an explanation, our approach can produce user-centered explanations that allow an individual to contest and act to change the decision to which they were subject [Wachter *et al.*, 2017b].

Our formulation allows an individual to inquire about any change to a decision that can be represented with a constraint set on the original formulation. With some restrictions, we show that counterfactual explanations for models with two common objective functions can be generated with limited additional computational effort beyond solving the original model. We believe this general approach to counterfactual explanations for optimization-based decisions is a fruitful yet nascent direction that deserves further development.

**Acknowledgements** This research was supported by the Natural Sciences and Engineering Research Council of Canada.

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
