# OpenReview forum: "Counterfactual Explanations for Optimization-Based Decisions in the Context of the GDPR"
_icaps-conference.org/ICAPS/2021/Workshop/XAIP — XAIP 2021_

### Official Review · AnonReviewer2 · 2021-07-06
**The authors use inverse combinatorial optimization to generate forms of counterfactual explanations to restricted versions of problems that have been expressed in specified ways. This is preliminary work.**

**Rating:** 4
**Confidence:** 4

**Review:**

The authors use inverse combinatorial optimization to generate forms of counterfactual explanations to problems that have been expressed in specified ways – WSBV minimisation (weighted sum of binary variables), and WSIV minimisation (weighted sum of integer variables).

Simplifying assumptions are made, to enable tractability. The assumptions are strong, and the usefulness to a human recipient of the generated explanations not fully demonstrated, and definitely not demonstrated in a realistic  “GDPR context” – so the title, at the very least, needs modification, and more completely worked examples would assist in persuading the reader that the restricted forms are sufficiently expressive of interesting problems worth explaining.

The key results Theorem 1 and Theorem 2 appear correct, though I did not fully check.

The proof of Theorem 1 is particularly straightforward because of the restriction ((1) of section 3.1) to one constraint on one component of x. The usefulness of this in realistic problem situations is questionable, and certainly hasn’t been adequately defended or demonstrated in the paper. The example of appointment scheduling is exceedingly unlikely, in practice, to be reducible to a single variable. And the claim that such a restriction affords a level of privacy to the other individuals involved would need to be evidenced by a full representation of a real-life problem, and solution context. For example, in the discussion at the top of page 4, could it not be the case that from awareness of how much my benefit can decrease before I lose access to the service, I could derive another’s benefit assessment?

Overall the work is somewhat preliminary, so many of the claims are too strong.

Given limited space, it is arguably more important to strengthen the motivation, and make space by compacting some of the proof steps - especially re Theorem 2 where the steps seem mechanical, rather than insightful.

In the related work section, for example, the work of Brandao and Magazzeni mentioned in the discussion (para 4) is *not* substantially narrower in scope than that of the present paper.

---

### Official Review · AnonReviewer3 · 2021-07-07
**A Preliminary Work but Is Definitely of Interest to the Workshop**

**Rating:** 7
**Confidence:** 4

**Review:**

The paper looks at the problem of generating counterfactual explanations for decisions generated from optimization problems. The paper argues that one could view such explanations through the lens of inverse optimization (or through an even more general framework of NCE). While the general problem of inverse optimization is too complex, the authors identify some specific problem settings where such solutions can be identified relatively easily. Some of the core restrictions they make are the fact that the alternative requested by the user strictly relates to a single component of the solution and the explanation generation process can only update a single parameter and additionally, this parameter can’t be part of the constraints. This means that the explanation generation process will not change the space of feasible solutions, but only change the preference between the different solutions. They present how, under such restrictions, explanations can be generated for the weighted sum of binary variables and for the weighted sum of non-negative integer variables (where the weights are also non-negative integer values).

I completely agree with the authors that inverse-optimization and even NCE presents clean mathematical formulations to study counterfactual explanations. My main problem with the paper is that the problems considered in the paper are too restrictive to generate useful explanations for practical problems. For example, one could compile certain flavors of cost-optimal planning into weighted MaxSat. In this case, let’s say the user’s questions are restricted to the form "Why not use an action a_1 at time-step t?" The explanation generated by the current method would basically involve reducing the cost of action a_1 until it catches up to the optimal solution. I would argue that this would be a pretty unsatisfactory explanation under most circumstances. For one, by forgoing updates of all other parts of the domain, depending on the setting, the cost of the actions would have to drop by ridiculous amounts to catch up to the optimal cost. This also limits the possibility of applying the techniques in debugging settings (though it may not be relevant in the context of GDPR). Moreover, if the user’s question is not stemming from confusion on the cost function of the action, just saying we need to reduce the action cost to make the plan work may not be convincing. This means that the method may need to be coupled with other techniques to create satisfactory explanations.

With that said, I still think the paper looks at an interesting topic and makes concrete contributions and a solid base to build on. Additionally works like (Brandao et al. 20) have shown that model reconciliation explanations could also be framed as an inverse optimization problem, so it would be interesting to see if any non-trivial classes of such explanations could also be efficiently calculated along the lines the authors have laid out in the paper. So I would argue for the paper to be accepted. Below I have provided some specific suggestions and also provided some questions that I would encourage the authors to respond to and incorporate into the paper.

Related Work: In my opinion, the closest related work in planning to the methods discussed in the paper are not the contrastive explanation works, which usually focus on identifying alternate solutions that meet the foil specification and analyzing the properties of such solutions. Rather I would imagine the current method is closer to works like [1] which tries to identify what they call excuses. In [1], given an unsolvable planning problem, the method tries to identify model updates (in their case changes to static variable values in the initial state) that would render the planning problem solvable. Obviously, their problem doesn’t fit the assumptions made by the authors, but I would argue that it is in the same spirit, and it is also interesting to note that this problem can be framed as a planning problem.


Minimality of Change: Another point to note that minimality of explanation is for the sake of the user. So what changes are more acceptable as compared to others may not be quite as obvious and may depend on the end-user. While for a single weight it might not matter as much, but as you try to extend the work to those that update multiple parameters, you may need to consider more complex distance functions, even those that may need to be learned.

Foils that are optimal: This is a smaller note. There could be genuine cases where the human may have a different objective function than the system and may ask for a foil that is as good as the current solution (so the foil is also optimal). I understand that there is no real explanation generation problem here, but there are many scenarios where such questions do arise and need to be handled. For example, in the case of cooperative decision-making like iterative planning [2], the system may need to change its decision to an equally good solution if the human reveals some additional preference (in the form of a foil constraint).

Clarification about the minimal C^0_j value: Since the \Delta function is monotonic wouldn’t the minimal value of  C^0_j be the same as the C^*_j value. If not could you add some discussion clarifying this, if so, the line in section 5.2 “The smallest possible value of c^0_j will be given by…” makes it sound like this value is somehow different from C^*_j.

One variable corresponds to one user: This seems extremely encoding specific to me, is there a reason to believe that this would hold in the majority of cases.

## Suggestions/Questions (Some questions and suggestions you might want to consider in future work)

1. Please specify some practical planning or scheduling settings where the current methods generate useful explanations. This could include cases where it is more realistic to change the parameters related to the objective function than any other constraints of the problem, at least from an end user’s point of view. Here the authors are free to interpret what it means for a change to be more realistic, for example maybe the effort needed to make that change or likelihood of making that change, etc. Also here I am making an assumption that the users would be more willing to accept counterfactuals that correspond to certain changes than others, maybe the authors disagree with this assumption

2. A lot of recent work in counterfactual explanations in ML focuses on identifying actionable counterexamples. For example, in the case of a loan rejection explanation, the system might want to give counterexamples that changes variables that are under the user’s control and they could realistically update. How would you bring in concepts like actionability into this framework?


[1] Göbelbecker, Moritz, et al. "Coming up with good excuses: What to do when no plan can be found." Twentieth International Conference on Automated Planning and Scheduling. 2010.

[2] Smith, David E. "Planning as an iterative process." Twenty-Sixth AAAI Conference on Artificial Intelligence. 2012.

---

### Official Review · AnonReviewer1 · 2021-07-08
**good paper that deserves to be presented at XAIP**

**Rating:** 7
**Confidence:** 3

**Review:**

Interesting paper that deserves to be presented at workshop

The article proposes a novel approach to a counterfactual explanation for optimization-based decisions using a generalization of inverse combinatorial optimization. Compared to the classic theory of counterfactual explanation (see. [Wachter et al., 2017b]), the inverse combinatorial optimization is aimed at finding the minimal change to the parameter values of the model. The authors show in detail how to formulate the inverse problem, including constraints coming from the GDPR regulation, and how to solve it under two common objective functions ((i) sum of weighted binary variables and (ii) sum of weighted integer variables.)
The paper is well written, the formalisation seems sound to me.

COMMENTS:

-- In some parts of Introduction and Background the concept of “Counterfactual Explanation” is confused with the concept of “Contrastive Explanation” or not correctly interpreted. Specifically:

	a. In 2.1: “First there is a counterfactual question, “Why decision P and not Q ?” this is the definition of Contrastive Explanation (see. Miller, T. 2019. Explanation in artificial intelligence: Insights from the social sciences. Artificial Intelligence 267)
	b. In 1: “In this paper, we adopt the approach of counterfactual explanations [Wachter et al., 2017b] by deriving the minimal change to the world that would have resulted in the alternative indicated by the questioner.” In the work of Watcher, the term world referred to the values of the variables while, in this work, refers to the coefficients associated with the variables.

-- In the conclusion, the authors claimed that “our approach can produce user-centred explanations that allow an individual to contest and act to change the decision to which they were subject”. This statement is not coherent with the approach, because the user cannot act to change the coefficient provided by the model, which is defined by the developer of the model.
-- Furthermore, the work relies on the assumption that “the users know and understand their weight”. This assumption is extremely strong and not applicable in all the contexts where the end-user is not technical.
-- The work is suitable only for simple models like the two presented, that are transparent-by-design. Finding a solution for inverse combinatorial optimization, considering complex models like neural networks, is extremely complex. furthermore, assuming it is possible to find it, it would not be interpretable.

---

### Meta-Review · Program_Chairs · 2021-07-08

**Recommendation:** Accept
**Confidence:** 5

**Metareview:**

The paper will be an interesting addition to the workshop proceedings. Please refer to the reviewers' comments on the limitations of the model and discuss with the PC on points you agree or disagree with (and incorporate revisions in the camera-ready accordingly). Overall, I do agree with Reviewer2's point on the somewhat weak connection to GDPR to make this a leading theme in the title. Also, some direct instantiations of the optimization problem for known planning problems would be a useful addition for this audience.

---

### Decision · Program_Chairs · 2021-07-08

Accept